# Decoupling regularization from the action space

## Abstract

Regularized reinforcement learning (RL), particularly the entropy-regularized kind, has gained traction in optimal control and inverse RL. While standard unregularized RL methods remain unaffected by changes in the number of actions, we show that it can severely impact their regularized counterparts. This paper demonstrates the importance of decoupling the regularizer from the action space: that is, to maintain a consistent level of regularization regardless of how many actions are involved to avoid over-regularization. Whereas the problem can be avoided by introducing a task-specific temperature parameter, it is often undesirable and cannot solve the problem when action spaces are state-dependent. In the state-dependent action context, different states with varying action spaces are regularized inconsistently. We introduce two solutions: a static temperature selection approach and a dynamic counterpart, universally applicable where this problem arises. Implementing these changes improves performance on the DeepMind control suite in static and dynamic temperature regimes and a biological sequence design task.

## 1 Introduction

Regularized reinforcement learning (RL) (Geist et al., 2019) has gained prominence as a widely-used framework for inverse RL (Rust, 1987; Ziebart et al., 2008; Fosgerau et al., 2013) and control (Todorov, 2006; Peters et al., 2010; Rawlik et al., 2012; Van Hoof et al., 2015; Fox et al., 2016; Nachum et al., 2017; Haarnoja et al., 2017; 2018; Garg et al., 2023). The added regularization can help with robustness (Derman et al., 2021), having a policy that has full support (Rust, 1987), and inducing a specific behavior (Todorov, 2006). However, we show that these methods are not robust to changes in the action space. We argue that changing the action space should not change the optimal regularized policy under the same change. For instance, changing the robot's acceleration unit from meters per second squared to feet per minute squared should not lead to a different optimal policy. While Haarnoja et al. (2018)'s heuristic is a step in the right direction, we argue that the heuristic does not reflect the structure of the action space, just the number of actions, and does not generalize to other regularizers MDPs.

The key idea proposed here is to control the range of the regularizer by changing the temperature. Indeed, by not changing the temperature, we demonstrate that we inadvertently regularize states with different action spaces differently. We show that for regularizers that we call standard, which include entropy, states with more actions are always regularized more than states with fewer actions. We introduce decoupled regularizers, a class of regularizers that fit Geist et al. (2019)'s formalism and have constant range. We show that we can convert any non-decoupled regularizer into a decoupled one.

Our contribution is as follows. First, we propose a static temperature selection scheme that works for a broad class of regularized Markov Decision Processes (MDPs), including entropy. Secondly, we introduce an easy-to-implement dynamic temperature heuristic applicable to all regularized MDPs. Finally, we show that our approach improves the performance on benchmarks such as the DeepMind control suite (Tassa et al., 2018) and the drug design MDP of Bengio et al. (2021).

## 2 PRELIMINARIES

A discounted MDP is a tuple $(\mathcal{S}, \mathcal{A}, \mathbb{A}, R, P, \gamma)$ where $\mathcal{S}$ represents the set of states, $\mathcal{A}$ is the collection of all possible actions and $\mathbb{A}(s)$ represents the set of valid actions at state $s$. If $|\mathbb{A}(s)|$ is not constant for all $s \in \mathcal{S}$, we say that the MDP has state-dependent actions. The reward function, denoted by $R : \mathcal{S} \times \mathcal{A} \to \mathbb{R}$ maps state-action pairs to real numbers. The transition function, $P : \mathcal{S} \times \mathcal{A} \to \Delta(\mathcal{S})$, determines the probability of transitioning to the next state, where $\Delta(\mathcal{S})$ indicates the probability simplex over the set of states $\mathcal{S}$. Additionally, the discount factor, represented by $\gamma \in (0, 1]$, is included in our problem formulation.

When solving a Markov Decision problem under the infinite horizon discounted setting, the aim is to find a policy $\pi(s) : \mathcal{S} \to \Delta(\mathbb{A}(s))$ that maximizes the expected discounted return $V_\pi(s) \triangleq \mathbb{E}\left[\sum_{t=0}^\infty \gamma^t R(s_t, A_t)|s_0 = s\right]$ for all states. A fundamental result in dynamic programming states that the value function $V_{\pi^\star}$ for any stationary optimal policy $\pi^\star$ must satisfy the Bellman equations (Bellman, 1954):

$$V(s) = \max_{\pi_s \in \Delta(\mathbb{A}(s))} \mathbb{E}_{a \sim \pi_s}[Q(s, a)] \quad \forall s \in \mathcal{S},$$

where $Q(s, a)$ is defined as $R(s, a) + \gamma \mathbb{E}_{s' \sim P(s,a)}[V(s')]$. Regularized MDPs (Geist et al., 2019) introduce a strictly convex regularizer $\Omega$ with temperature $\tau$ to regularize the policy as

$$V(s) = \max_{\pi_s \in \Delta(\mathbb{A}(s))} \mathbb{E}_{a \sim \pi_s}[Q(s, a)] - \tau \Omega(\pi_s) = \Omega_\tau^\star(Q(s, \cdot)),$$

such that $V_\pi(s) \triangleq \mathbb{E}\left[\sum_{t=0}^\infty \gamma^t (R(S_t, A_t) - \tau \Omega(\pi(\cdot|s)))|S_0 = s\right]$. The optimal policy equals the gradient of $\Omega_\tau^\star$ (Geist et al., 2019). Replacing $\Omega$ by the negative entropy yields soft Q-learning (SQL) as $\Omega_\tau^\star$ is the log-sum-exp function at temperature $\tau$ ($\tau \log \sum_a \exp(Q(s, a)/\tau)$) and $\nabla \Omega_\tau^\star$ is the softmax at temperature $\tau$, $\pi(a|s) \propto \exp(Q(s, a)/\tau)$.

Whereas the proposed approach applies to all regularized MDPs, we focus on the case where $-\Omega$ is the entropy. There are two main reasons for this choice. First, it is widely used (e.g. Ziebart et al., 2008; Haarnoja et al., 2017). Second, it allows us to derive analytical bounds. Other alternatives include Tsallis entropy (Lee et al., 2019).

## 3 GRAVITATION TOWARDS REGULARIZATION

To quantify the impact of a change in action space on regularization, we first define the range of a regularizer.

**Definition 1.** The range of the regularizer $\Omega$ over the action space A, $L(\Omega, A)$ is $\sup_{\pi \in \Delta(A)} \Omega(\pi) - \min_{\pi \in \Delta(A)} \Omega(\pi)$.

The range is sometimes used for analyzing regularized follow-the-leader algorithms (e.g., Theorem 5.2 Hazan et al., 2016), and its square is referred to as the *diameter* (Hazan et al., 2016). If the range depends on the action space of the state, the propagation of the regularization by the Bellman equation can have a compounding effect. Thus, the change in regularization affects not only the state itself but all states that can reach it. Thus, balancing regularization and reward maximization in MDPs in sequential decision-making processes is crucial. We show this using two small illustrative examples.

*Example* 1. (Bias due to $|\mathbb{A}(s)|$) In the MDP shown in Figure 1a, with reward $r$ on all transitions starting at $s_1$ and zero otherwise, the probability of taking the action $a_0$ is $\frac{1}{n+1}$ (where $n + 1$ is the number of paths) with no discounting.

*Proof.* The result follows from the definition of V. The value of $s_2$ is $\tau \log n$, thus the probability of taking the action $a_0$ is $\frac{\exp r/\tau}{\exp r/\tau + \exp (r + \tau \log n)/\tau}$ at temperature $\tau$. $\square$

*Example* 2. (Bias due to loops) In the MDP shown in Figure 1b, with reward $r$ on all transitions, the probability of taking the action $a_0$ is $1 - n \exp(r/\tau)$, at temperature $\tau$ and no discounting.

*Proof.* The value of $s_1$, $V$, equals $\tau \log[n \exp(r/\tau + V/\tau) + \exp(r/\tau)]$ or $r - \tau \log(1 - n \exp(r/\tau))$. Thus, the probability of taking $a_0$ is $\exp(r/\tau)/\exp(V/\tau) = 1 - n \exp(r/\tau)$. The MDP diverges if $1 \leq n \exp(r/\tau)$ $\square$

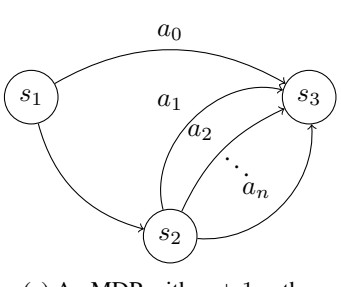

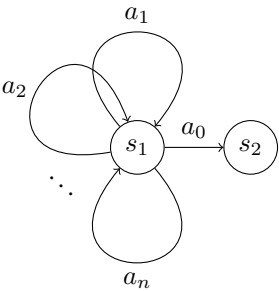

(a) An MDP with $n + 1$ paths.

(b) An MDP with $n$ loops.

Figure 1: Toy MDPs

These examples show a gravitation towards regularization. Concretely, with negative entropy, the regularization at the states with larger action spaces is greater, resulting in a higher regularization and higher reward to pass through those states. Thus, when we increase $n$, the probability of passing through the state with $n$ or $n + 1$ actions increases. However, the states should be regularized consistently, and how much we regularize a state should not depend on its action space. One way of measuring this quantity is the range defined in Definition 1. Indeed, we argue that the range should not depend on the action space. This motivates our solution, which we call decoupled regularizers.

Despite the specificity of these examples, the same behavior can be observed more broadly with other regularizers, including those in stochastic MDPs (Mai and Jaillet, 2020). To this end, we introduce a general class of regularized MDPs that show a similar problem in Section 5. It is also important to note that the discount factor was set to one for mathematical clarity, and including discounting alleviates the risk of divergence but does not completely eliminate the problematic behavior. In the following, we introduce our approach to address inconsistent regularization across action spaces.

## 4 DECOUPLED REGULARIZERS

We note that in the following we can replace the sum with an integral in the continuous actions space. We look at differential entropy and continuous actions in Section 7.

**Definition 2.** We call a regularizer $\Omega$ decoupled if the range of $\Omega$ is constant over all action spaces $\mathbb{A}(s)$ for all valid states $s$. For any non-decoupled regularizer $\hat{\Omega}$ at state $s$, $\Omega(\pi)$, defined as $\hat{\Omega}(\pi)/L(\hat{\Omega}, \mathbb{A}(s))$ is the decoupled version of $\hat{\Omega}$

Concretely, the value of a regularized MDP at state $s$ is given by

$$V(s) = \Omega_\tau^\star(Q(s, \cdot)), \tag{1}$$

which we propose to replace with

$$V(s) = \Omega_{\tau/|L(\Omega, \mathbb{A}(s))|}^\star(Q(s, \cdot)). \tag{2}$$

We give the range of some commonly used regularizers on discrete actions in Table 1. Note that in the Tsallis case, $q$ is often set to 2. While there are no known analytical solutions for the convex conjugate of Tsallis entropy, when $q = 2$, it can be solved efficiently (Michelot, 1986; Hazan et al., 2016; Duchi et al., 2008). We further note that the convex conjugate of KL with the uniform distribution (denoted $U$) is sometimes called mellowmax (Asadi and Littman, 2017). The relationship between mellowmax and KL divergence was first shown in Geist et al. (2019). The range for the negative entropy regularizer is $\log |\mathbb{A}(s)|$, which equals the logarithm of the number of actions. Thus the effective temperature is $\tau/\log |\mathbb{A}(s)|$ as the minimum discrete entropy is zero. Entropy divided by maximum entropy is called efficiency (Alencar, 2014).

## 5 STANDARD REGULARIZERS AND THE DRIFT IN RANGE

We now look at a general class of regularizers over discrete action spaces.

|  | $H(\pi)$ | $\mathrm{KL}(\pi\|U)$ | negative Tsallis entropy |
|---|---|---|---|
| $\Omega(\pi)$ | $\sum_{a\in\mathbb{A}(s)}\pi(a\|s)\log\pi(a\|s)$ | $\sum_{a\in\mathbb{A}(s)}\pi(a\|s)\log\left(\pi(a\|s)/\frac{1}{\|\mathbb{A}(s)\|}\right)$ | $\frac{k}{q-1}\left(\sum_{a\in\mathbb{A}(s)}\pi(a\|s)^q-1\right)$ |
| $\Omega^{\star}(Q(s,\cdot))$ | $\tau\log\sum_{a\in\mathbb{A}(s)}\exp(Q(s,a)/\tau)$ | $\tau\log\left(\frac{1}{\|\mathbb{A}(s)\|}\sum_{a\in\mathbb{A}(s)}\exp(Q(s,a)/\tau)\right)$ |  |
| $\nabla\Omega^{\star}_{\tau}((s,\cdot))$ | $\dfrac{\exp(Q(s,a)/\tau)}{\sum_{a\in\mathbb{A}(s)}\exp(Q(s,a)/\tau)}$ | $\dfrac{\exp(Q(s,a)/\tau)}{\sum_{a\in\mathbb{A}(s)}\exp(Q(s,a)/\tau)}$ |  |
| $\sup_{\pi\in\Delta(\mathbb{A}(s))}\Omega(\pi)$ | $0$ | $\log\|\mathbb{A}(s)\|$ | $0$ |
| $\min_{\pi\in\Delta(\mathbb{A}(s))}\Omega(\pi)$ | $-\log\|\mathbb{A}(s)\|$ | $0$ | $\frac{k}{q-1}\left(\frac{1}{\|\mathbb{A}(s)\|^q}-1\right)$ |
| $L(\Omega,\mathbb{A}(s))$ | $\log\|\mathbb{A}(s)\|$ | $\log\|\mathbb{A}(s)\|$ | $\frac{k}{q-1}\left(1-\frac{1}{\|\mathbb{A}(s)\|^q}\right)$ |

Table 1: Different values at state $s$. Empty cells indicate no known analytical solution.

**Definition 3.** We call a regularizer in the form $\Omega(\pi_s) = g(\sum_a f(\pi_s(a)))$ for a strictly convex function $f$ and a strictly monotonically increasing function $g$ a standard regularizer. We assume that $\Omega(\pi)$ is strictly convex to be compatible with regularized MDPs.

We chose this form because it is easy to reason about yet general enough to encapsulate many regularizers, including entropy and Tsallis entropy. The regularizer is invariant to permutation and naturally extends itself to higher dimensions.

In addition, we include one regularity assumption.

**Assumption 1.** (Symmetry) we assume that $f(0)$ and $f(1)$ are equal to $0$.

We now show that under Assumption 1 the supremum is constant.

**Lemma 1.** *Under Assumption 1, the supremum of the regularizer is equal to the limit of the regularizer at a deterministic distribution (i.e., only one action has non-zero probability, and the others have zero probability).*

*Proof.* The supremum of $f$ is at $0$ and $1$. By convexity, at any other point between $0$ and $1$, $f$ is smaller than $0$. By strict convexity $\forall_{0<p<1} f(p) < pf(1) + (1-p)f(0) = 0$ $\square$

Assumption 1 helps us identify the supremum of the regularizer and show that it is constant. However, it is possible that Lemma 1 holds even if Assumption 1 does not. For instance, the maximum negative Tsallis entropy is always zero.

With the supremum being $g(0)$, we now calculate the minimum regularizer.

**Lemma 2.** *The minimum regularization is achieved under the uniform policy.*

*Proof.* We proceed with a proof by contradiction. Suppose that the minimum is a nonuniform policy. Since the uniform policy is in the convex hull of all permutations of that policy, by strict convexity and symmetry, it has a lower value than the supposed minimum and thus is contradictory. $\square$

We now show that the minimum of the regularizer, attained at the uniform policy, is decreasing in the number of actions.

**Lemma 3.** *The minimum regularization decreases with the number of actions.*

*Proof.* By strict convexity, we have that $f(0) > f(x) + f'(x)(0-x)$. By Assumption 1, we have that $xf'(x) - f(x)$ and as a consequence $f'(x)/x - f(x)/x^2$, the gradient of $f(x)/x$, is always positive. This implies that minimum regularization decreases as $x = 1/n$ decreases. $\square$

Equipped with these three lemmas, we can now show that the range grows with the number of actions.

**Theorem 1.** *The range of standard regularizers grows with the number of actions.*

*Proof.* While the minimum grows as per Lemma 3, the supermum stays the same per Lemma 1, and the range hence grows with the number of actions. □

*Remark* 1. This result holds for any regularizer invariant to permutation, and its supremum is constant. The standard form guarantees permutation invariance. For instance, the range of negative Tsallis entropy also grows with the number of actions.

*Remark* 2. We have made no comment on the rate at which the range grows. For instance, the range of Tsallis entropy grows to 1, and thus, the range of negative Tsallis entropy does not grow as fast as negative entropy with respect to the number of action spaces.

## 6 VISITING DECOUPLED MAXIMUM ENTROPY RL

In this section, we review decoupled maximum entropy RL, revisit the examples provided in Section 3, and show that decoupling improves the convergence of undiscounted entropy regularized MDPs.

---
**Algorithm 1** Decoupled SQL
---
Sample $s$ from $\mathbb{P}^0$
**if** decoupled **then**
$\quad \tau' \leftarrow \tau/\log|\mathbb{A}(s)|$
**else**
$\quad \tau' \leftarrow \tau$
**end if**
**while** true **do**
$\quad$ Sample action $a \in \mathbb{A}(s)$ with probability $\exp(Q(s,a)/\tau')/\sum_{a' \in \mathbb{A}(s)} \exp(Q(s,a')/\tau')$
$\quad$ Play action $a$ and observe $s', r$
$\quad$ **if** decoupled **then**
$\quad\quad \tau' \leftarrow \tau/\log|\mathbb{A}(s')|$
$\quad$ **end if**
$\quad Q(s,a) \leftarrow r + \tau' \log \sum_{a' \in \mathbb{A}(s')} \exp(Q(s',a')/\tau')$
$\quad s \leftarrow s'$
**end while**

---

First, we provide an example of a tabular implementation in Algorithm 1. The conditional shows the changes needed to decouple SQL.

Next, we revisit the MDP in Figure 1a. Using decoupled SQL, we get that the probability of action $a_0$ constant as the value of $s_2$ is $\tau/\log n \log(n \exp(0 \log n/\tau) = \tau$ when regularizing by decoupled entropy. The value of $s_1$ is $\tau/\log 2 \log [\exp(r \log 2/\tau) + \exp(r \log 2/\tau + \tau \log 2/\tau)]$ or $r + \tau/\log 2 \log 3$. The probability of taking action $a_0$ is equal to $\exp(r \log 2/\tau - V(s_1) \log 2/\tau)$ or $1/3$.

The state $s_1$ of the MDP in Figure 1b will be at temperature $\tau/\log(n+1)$; thus, if $r$ is less than $-1$, it will not diverge. The probability of taking the action $a_0$ is $1 - n \exp(r \log n/\tau)$, which is strictly decreasing in $r$ and has a root at $r$ equal to $-1$; thus, the regularized Bellman equation converges below that threshold. The improved convergence of the MDP in Figure 1b using decoupled entropy motivates the following more general result.

**Proposition 1.** *In maximum entropy undiscounted inverse reinforcement learning with deterministic dynamics like Ziebart et al. (2008) or Fosgerau et al. (2013), decoupled entropy guarantees convergence if the maximum reward is less than $-\tau$.*

*Proof.* If $\sum_{a \in \mathbb{A}(s)} \exp R(s,a)/\tau < 1$, a solution always exists (Mai and Frejinger, 2022, Remark 2). Since $\exp(R(s,a) \log n/\tau) < 1/n$, the model always has a solution. □

## 7 AUTOMATIC TEMPERATURE FOR REGULARIZED MDPS

Haarnoja et al. (2018) proposed adding a lower bound on the entropy of the policy to find the right temperature. Concretely, they propose using the constraint $H(\pi(\cdot|s)) \geq \bar{H}(\mathbb{A}(s))$ for some function

$\bar{H}$ to the Bellman equation. They propose using the dual of the aforementioned constraint as the temperature leading to Algorithm 2. We note that we parametrize $\tau$, the dual variable and temperature, in terms of its logarithm so that it stays positive. While this change deviates from Haarnoja et al. (2018)'s notation, it better reflects their actual implementation.

---

**Algorithm 2** Soft actor-critic's update

---

$s$ is sampled from $\mathbb{P}^0$
**while** 1 **do**
    $a$ is sampled from $\pi(\cdot|s)$
    $s'$ is sampled by playing $a$
    $\theta \leftarrow \theta - \lambda \nabla_\theta J_Q(\theta)$                           $\triangleright$ Update critic (3a)
    $\phi \leftarrow \phi - \lambda \nabla_\phi J_\pi(\phi; \tau)$                      $\triangleright$ Update policy (3b)
    $\log \tau \leftarrow \log \tau + \lambda \nabla_{\log \tau} J_{\log \tau}(\log \tau; \bar{H}(\mathbb{A}(s)))$    $\triangleright$ Update temperature (3c)
    $s \leftarrow s'$
**end while**

---

$$J_Q(\theta) = \mathbb{E}_{a' \sim \pi(\cdot|s')}[(Q_\theta(s, a) - (r + \gamma Q_\theta(s', a') - \tau \log \pi_\phi(a'|s')))] \tag{3a}$$

$$J_\pi(\theta) = \mathbb{E}_{a \sim \pi(\cdot|s)}[\tau \log \pi_\phi(a|s) - Q_\theta(s, a)] \tag{3b}$$

$$J_{\log \tau}(\log \tau; \mathbb{A}(s)) = \tau \mathbb{E}_{a \sim \pi(\cdot|s)}[-\log \pi - \bar{H}(\mathbb{A}(s))] \tag{3c}$$

Haarnoja et al. (2018) propose using the negative dimensions of the actions as the target entropy. So if the action is a vector in $\mathbb{R}^n$, $\bar{H}$ is $-n$. Their proposed solution has two downsides: First, there is no reason that the same heuristic would be meaningful if another regularizer, for instance, if Tsallis entropy, was used. Second, Haarnoja et al. (2018)'s heuristic does not reflect the action space. Both of these points are easy to illustrate; if the action space is a real number from -5e-3 to 5e-3, the maximum entropy is -2, which would be lower than Haarnoja et al. (2018)'s heuristic, and make the problem infeasible. It is important to stress that $\tau$ will grow to infinity if the target entropy $\bar{H}$ is infeasible.

To remedy these, we propose a $\bar{H}$ inspired by the range of the regularizer $\Omega$. Concretely, we argue that

$$\Omega(\pi(\cdot|s)) - \sup_{\pi' \in \Delta(\mathbb{A}(s))} \Omega(\pi') \leq -\alpha L(\Omega; \mathbb{A}(s)) \tag{4}$$

should hold for some constant $\alpha$ between 0 and 1. Setting $\alpha$ to 0 is equivalent to disabling the constraint, and setting $\alpha$ to one results in $\pi$ becoming the minimum regularized (or maximum entropy) policy. Translating (4) back to entropy yields

$$H(\pi(\cdot|s)) \geq \alpha H(U) + (1 - \alpha)H(V), \tag{5}$$

where $U$ is the uniform and $V$ is the minimum reasonable entropy policy a policy should have. We need to define $V$ as the minimum entropy policy as it is not defined for differential entropy. We note again that setting $\alpha$ to one yields the uniform and $\alpha$ to zero yields the minimum entropy policy. Note that $H(U)$ is the logarithm of the volume of the action space, i.e., the logarithm of the integral of the unit function over the action space. We discuss choosing $\alpha$ in the next section.

## 8 EXPERIMENTS

In this section, we provide three sets of experiments: a toy MDP where the number of actions is a parameter, a set of experiments on the DeepMind Control suite (Tassa et al., 2018), and lastly, the drug design MDP of Bengio et al. (2021).

### 8.1 A TOY MDP

To illustrate the importance of temperature normalization, we propose a toy MDP where the number of actions is a parameter. The state $s$ is an $n$ dimension vector in the natural non-zero numbers

such that $s_i \leq m$ for all $i \in \{1, \ldots, n\}$ for some $m$. Each action increases or decreases one of the elements of $s$. Doing an action that would invalidate the state (for instance, make elements of $s$ zero) does not change the state. The agent starts at state $[1, 1, \ldots, 1]$. The agent receives a $-1$ reward for every time step that it has not reached the goal point $[m, m, \ldots, m]$. The episode terminates after reaching the goal point. When $n = 2$, the MDP is a grid where the agent starts a the bottom left corner and receives a negative reward as long as it has not reached the top right corner. The agent can only move to neighboring states but not diagonal ones. We display the expected time to exit with $\gamma = 0.99$ and $\tau = 0.4$ in Figure 2. The time to exit of SQL becomes very large for $n > 6$. It is important to stress that if the temperature is less than 1, decoupled SQL cannot diverge by Proposition 1. We also note that we have to set the temperature very low so that the SQL does not diverge at five dimensions, and thus, the decoupled version is fairly close to the shortest path. This example illustrates two main points: First, it highlights the importance of decoupling regularizers across benchmarks. Indeed, setting a unique temperature for all $n$ yields suboptimal behavior as the agent gains more regularization in higher dimension spaces, and the balance between reward and regularization is broken. Second, it highlights the improved convergence properties of decoupled entropy.

## 8.2 DEEPMIND CONTROL

The maximum entropy in the DeepMind control (DMC) benchmark is $n \log(2\beta)$, where $n$ is the number of actions and $\beta$ is chosen such that the actions are in the $-\beta$ to $\beta$ range. In the first experiment, we fix the temperature to $0.25$. The test rewards over the training, shown in Figure 3, do not worsen and can, in many instances, improve compared to the non-decoupled version. We provide the full experiments in Appendix B. While the model is sensitive to changes in action space, the gain in performance can still be observed across different values of $\beta$. In addition to analyzing how changes in reward scale change the performance as Henderson et al. (2018) suggests, we argue that it is also important to analyze how the performance

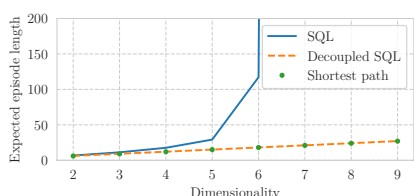

Figure 2: Episode length at different dimensions of the hypergrid problem.

changes in response to changes to the action space and range. We note that we do not use any scale invariant optimizer or loss function and that reaching full invariance to changes in action scale is beyond the scope of this work.

We now focus on the dynamic temperature setting. We chose $\alpha \approx 0.77$ to get similar results as Haarnoja et al. (2018) when the actions are in the $[-1, 1]$ range, this is our recommended default. Otherwise, the alternative is finding the optimal $\alpha$ as one would with the temperature as the interpretation is similar, the higher $\alpha$, the higher the final temperature will be.

As shown in Figure 4b, Haarnoja et al. (2017)'s heuristic becomes infeasible, leading to very high temperatures. High temperatures, in turn, lead to learning failure. Figure 4c shows similar performance as the temperature is extremely low for both models.

## 8.3 COMPARISON WITH GENERATIVE FLOW NETWORKS

Our final experiment involves the drug design by fragments of Bengio et al. (2021). In this MDP, an agent adds fragments, collections of atoms, to other fragments to build a molecule (we refer to Jin et al., 2018, for a more detailed description of the representation). The agent can end the episode when the state is a valid molecule, making the horizon finite but random. Each molecule is represented as a tree, and each fragment is a node in this tree. Each tree corresponds to a unique and valid molecule. GFlowNets (GFN) aims to sample molecules proportionally to some proxy that predicts reactivity with some material (Bengio et al., 2021). The goal is not to find only one molecule with a high reward but a diverse set of molecules with high rewards. As such, our main metric, other than high reward, is the number of modes or molecules that have a low similarity to other modes. We find the set of modes by iterating over the list of generated molecules and adding molecules that are not similar to any existing mode to that set.

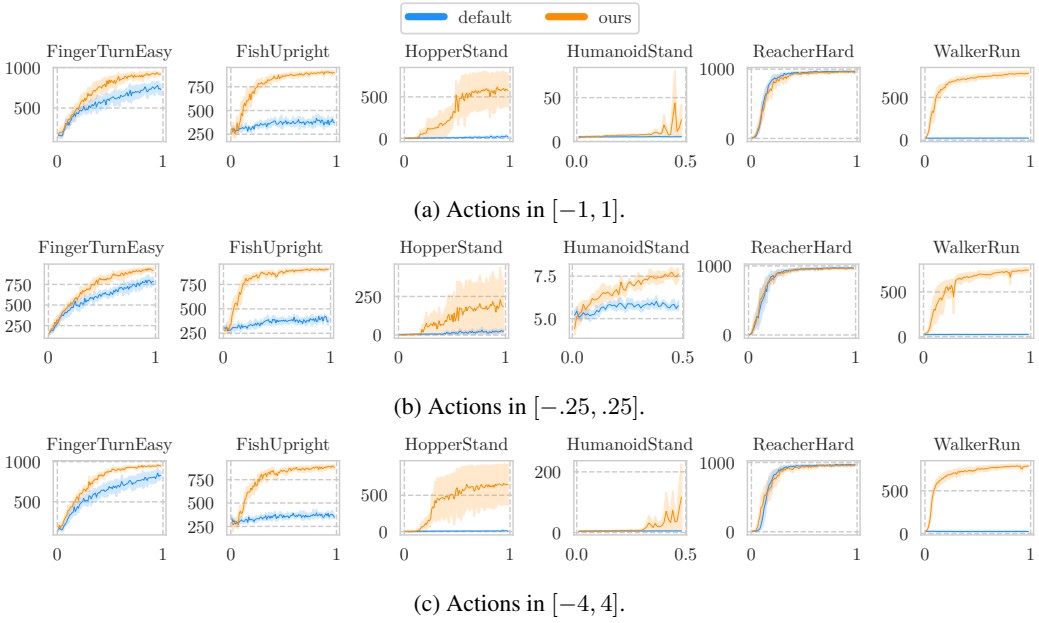

Figure 3: Test reward on the DMC benchmark with $\tau = 0.25$. The X axis is the number of iterations divided by $1e6$.

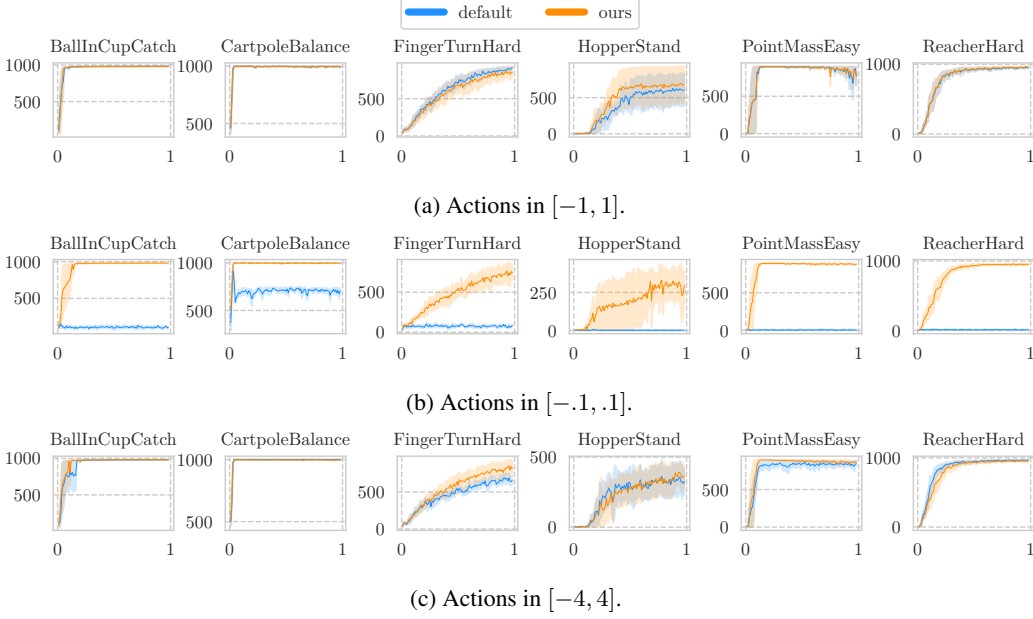

Figure 4: Test reward on the DMC benchmark with automatic temperature. The x-axis is the number of iterations divided by $1e6$.

It is beyond the scope of this work to properly introduce GFNs; we therefore simply state that they train policies to sample terminal states in proportion to an unnormalized distribution. Bengio et al. (2021) imposes four constraints on the MDP: there should only be one initial state, each state is reachable from the initial states, no state is reachable from itself, and the transition function is deterministic. Lastly, Bengio et al. (2021) assumes knowledge about the inverse dynamics. Concretely, for every state $s'$ they assume they have the list of all states $s$ that can reach $s'$, i.e., $\{s|\exists a \in \mathbb{A}(s)\text{s.t.}\mathbb{P}(s'|s, a) > 0\}$. These assumptions are not always easy to satisfy. For instance,

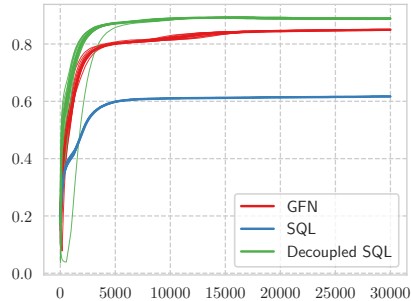 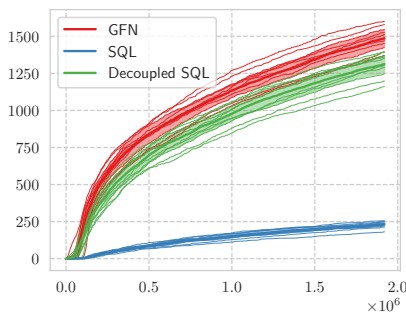

Figure 5: The left plot is the median reward of each batch. The right is the number of modes found. The shaded area shows the interquartile range, and the heavy line shows the interquartile mean.

undoing actions is not trivially possible with these assumptions. We use trajectory balance for the GFN loss (Malkin et al., 2022). For algorithm parity, we use path consistency learning (Nachum et al., 2017) as our SQL loss. We note that we use a static temperature.

The results in Figure 5 show the median reward and number of modes found. Indeed, the median reward of decoupled SQL is higher than SQL and GFN through training. The left subplot shows that decoupled SQL finds many high-quality modes. While SQL over-regularizes states with more actions, leading to a policy that prefers to pass through these hub states with many actions, decoupled SQL does not have this problem. This result alone highlights the need for decoupling in the state-dependent action setting.

## 9 CONCLUSION

In this paper, we argued that the amount we regularize should not depend on the action space. For example, we should not have to change the temperature of our regularized MDPs when we change the units of our robots. To illustrate our point, we introduced standard regularizers, which include entropy. We showed that standard regularizers increase how much they regularize with the number of actions. We proposed that the range should not depend on the action space and introduced decoupled regularizers as regularizers whose range is constant. We showed that we can obtain decoupled regularizers from normal regularizers by dividing them by their range. While instead of decoupling, we can change the temperature manually, we argue that it is often not desirable for benchmarks and cannot solve the problem in the state-dependent action setting. We emphasize the broad applicability of our findings; both the static and dynamic temperature schemes work for all regularized MDPs.

Perhaps most notably, our research has achieved unprecedented results in the domain of drug design. This is especially significant as Bengio et al. (2021) did not include SQL in their results as they mentioned it was too unstable and inherently prefers larger molecules. However, we found that our decoupled regularizers with PCL resolved both issues, serving as the critical, missing component. The innate simplicity of SQL, adaptability in environments characterized by cycles, and independence from inverse dynamics, the need to know which states can reach another state that is fundamental to GFNs, accentuate its appeal, and underscore its suitability for MDPs.

Our proposed method works regardless of the chosen regularizers but we only justified its use for *standard* regularizers; of course, not all regularizers are standard. For instance, $\pi^\top A \pi$ for strictly positive definite matrix $A$ is only standard if $A$ is the identity matrix. We posit that the same approach we took here might be insightful in that there may exist a function similar to the range that should be kept constant by transforming the regularizer. Lastly, while we moved closer to regularized scale-independent RL by introducing regularized RL models that are not sensitive to changes in action space, we believe there is more work to be done on the optimization side of the problem to enhance the scale invariance properties further.

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

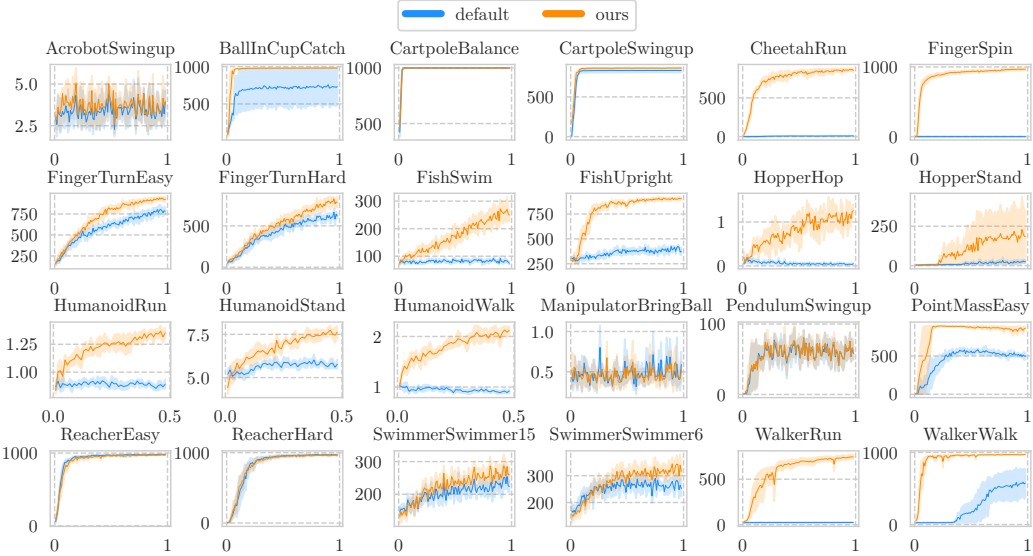

Figure 6: DMC test reward with the action scale set to 0.25. Static temperature.

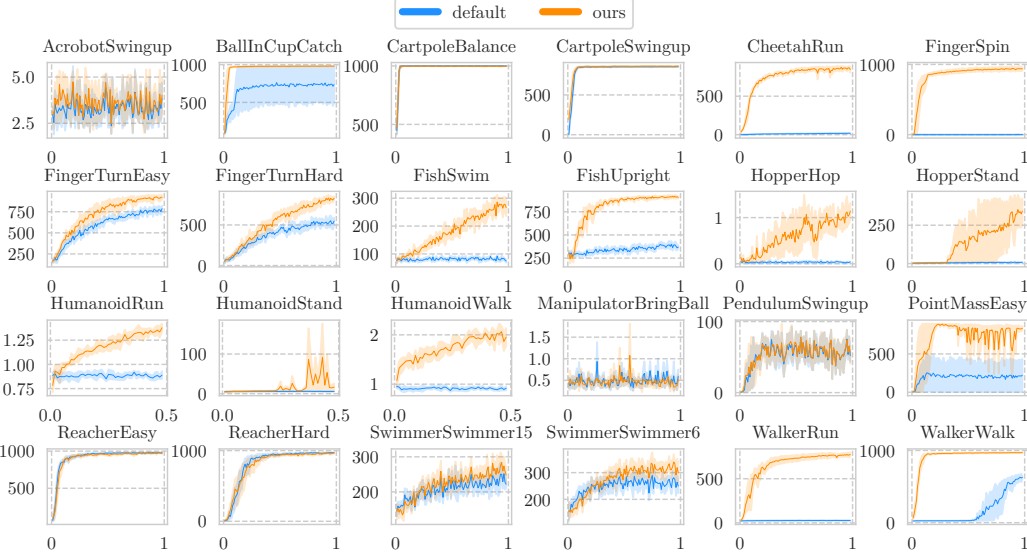

Figure 7: DMC test reward with the action scale set to 0.5. Static temperature.

## A   REPRODUCIBLITY

All code is hosted at `https://anonymous.4open.science/r/decoupled_sql-5CAB/` and `https://anonymous.4open.science/r/decoupled_gfn-8589`.

## B   EXTENDED DEEPMIND CONTROL EXPERIMENTS

We define our minimum entropy distribution $V$ as a uniform distribution over a $1e - 3$ range. We argue that for practical purposes, any distribution with such a low is deterministic for all intents and purposes.

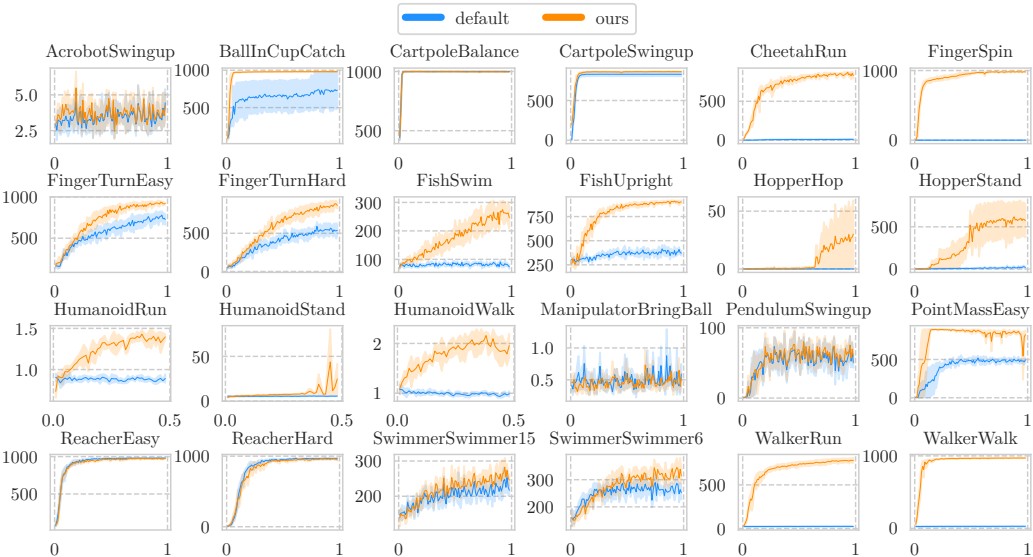

Figure 8: DMC test reward with the action scale set to 1. Static temperature.

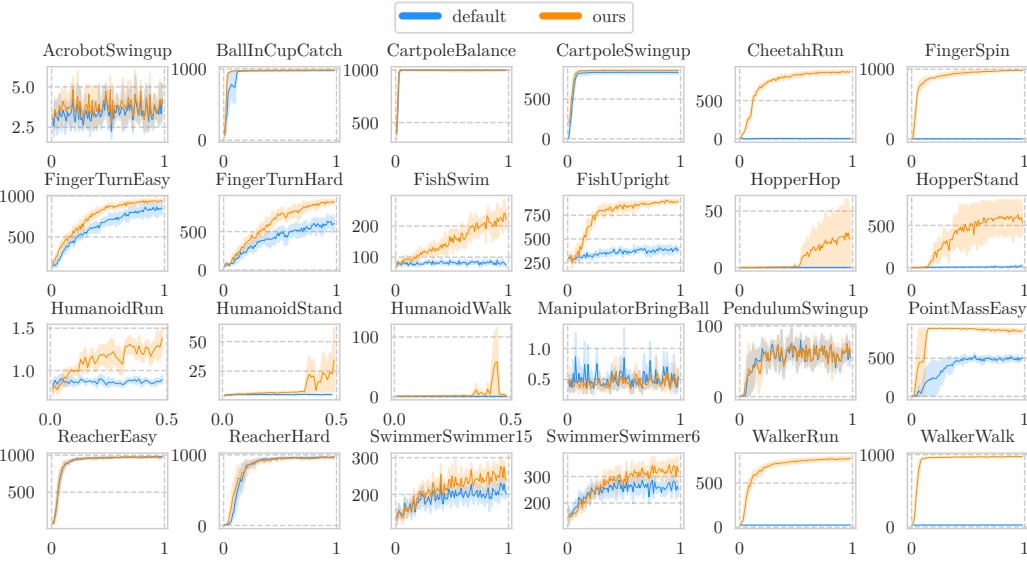

Figure 9: DMC test reward with the action scale set to 2. Static temperature.

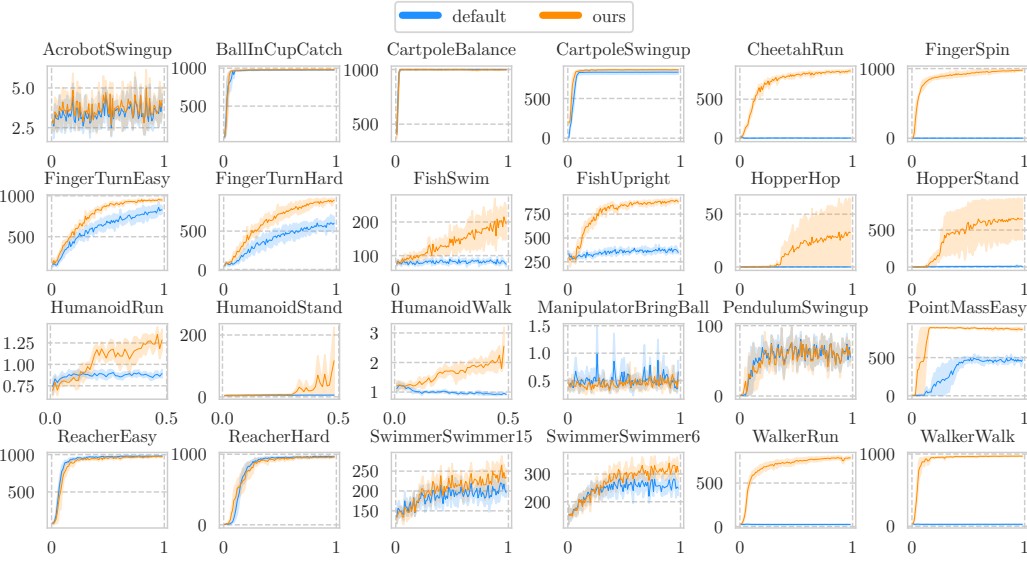

Figure 10: DMC test reward with the action scale set to 4. Static temperature.

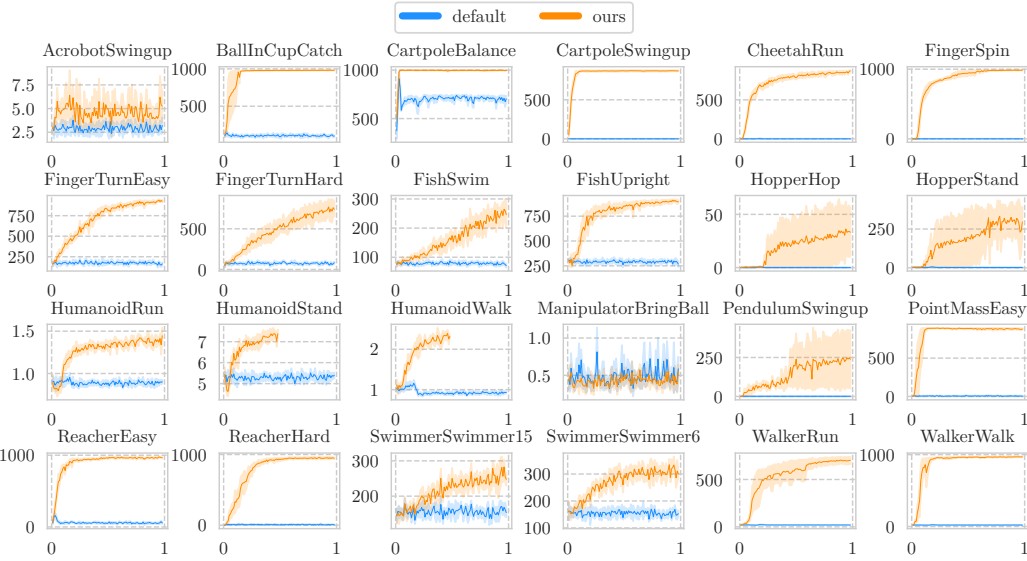

Figure 11: DMC test reward with the action scale set to .1. Dynamic temperature.

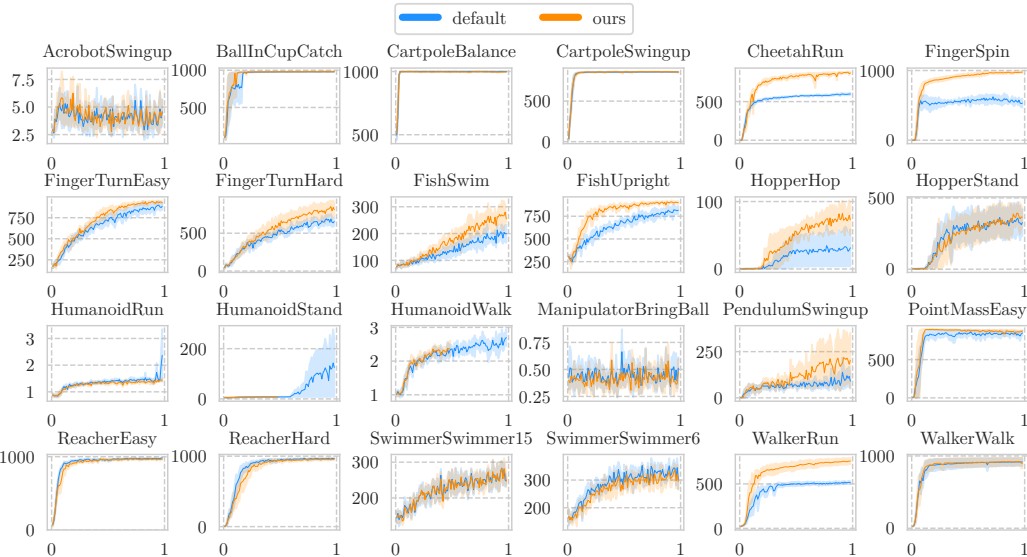

Figure 12: DMC test reward with the action scale set to .25. Dynamic temperature.

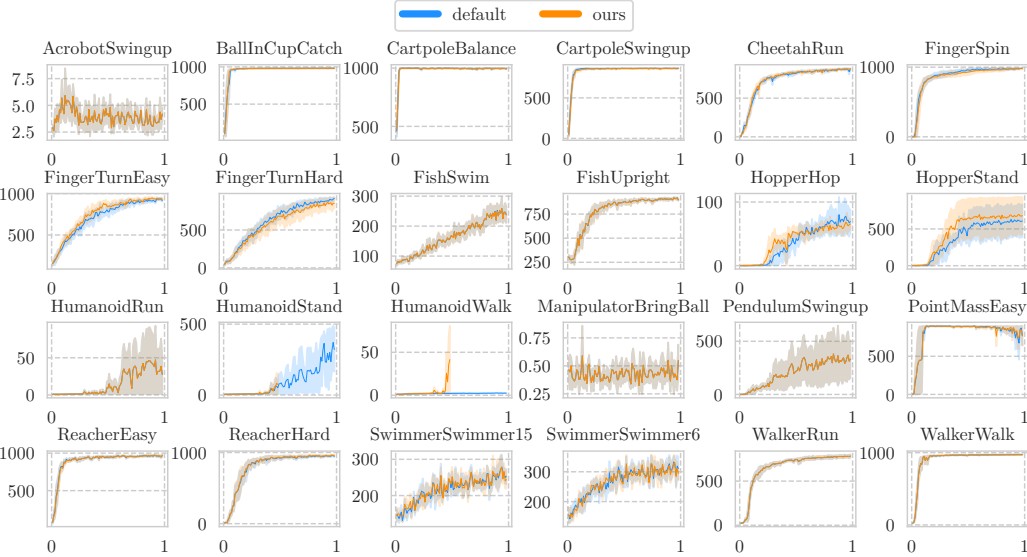

Figure 13: DMC test reward with the action scale set to 1. Dynamic temperature.

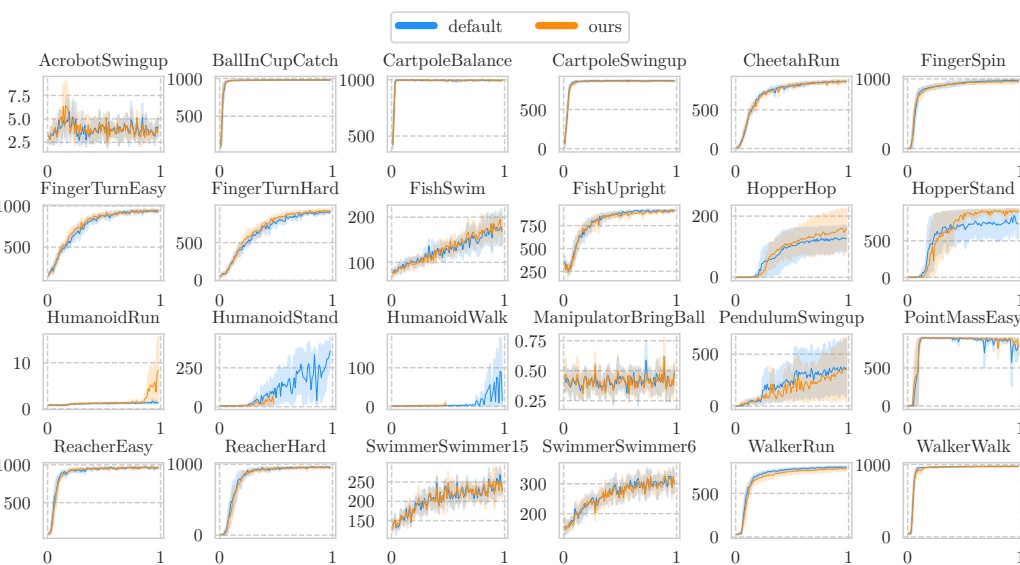

Figure 14: DMC test reward with the action scale set to 4. Dynamic temperature.

