# OpenReview forum: "Decoupling regularization from the action space"
_ICLR.cc/2024/Conference — ICLR 2024 poster_

### Official Review · Reviewer_g8g2 · 2023-11-08

**Soundness:** 3 good
**Presentation:** 3 good
**Contribution:** 3 good
**Rating:** 6
**Confidence:** 3

**Summary:**

This work reveals the problem that the regularization effect can be severely affected by the number of actions (especially when action spaces are state-dependent) in regularized reinforcement learning. Starting with two motivating examples, this paper introduces decoupled regularizers whose range is constant over action spaces. The authors further prove that the range of a general class of standard regularizers grows with the number of actions. Two solutions of temperature selection, static v.s., dynamic, are proposed based on which practical algorithms can be implemented. The proposed solutions are evaluated in toy examples, DMC and drug design environments with SQL, SAC and GFN as backbones and baselines. The results demonstrating the effectiveness of the proposed solutions.

**Strengths:**

- The paper is overall well organized and written. Illustrative examples and discussions are used for smooth reading.
- The proposed solutions to decoupling regularization and temperature adjustment are of generality to a family of commonly seen regularizations.
- The experiments cover toy examples, popular DMC environments and drug design problem. Noticeably, unprecedented results in the domain of drug design are achieved.

**Weaknesses:**

- I think the key hyperparameter $\alpha$ needs more discussion. It will be helpful to analyze the (empirical) effects of different choices of the value of $\alpha$ and recommend the values or the strategy of value selection.
- The content in the experiment part lacks of sufficient details, which is also missing in the appendix. I recommend the authors to add the key details of experiments.

**Questions:**

1) Is it possible to have some case-analysis for the results of DMC (e.g., in Figure 3,4)? For example, the proposed method achieves significantly better performance in tasks like BallInCupCatch, FingerTurnHard, HopperStand while the baseline fails totally.

2) Which is the type of temperature setting adopted for the experiments in Section 8.3, i.e., fixed temperature or dynamic temperature?

&nbsp;

Minor:
- It seems that the coefficients $\gamma$ and $\tau$ are missing for the entropy term in Equation 3a.

---

> ### Author Response · Authors · 2023-11-16
>
> We are grateful for your time on our work and the report. We provide answers the comments below.
>
> > I think the key hyperparameter $\alpha$ needs more discussion. It will be helpful to analyze the (empirical) effects of different choices of the value of $\alpha$ and recommend the values or the strategy of value selection.
> >
>
> We will include an ablation for $\alpha$. However, it might not be ready before the end of the discussion period.
>
> We added a small note in the paper that we recommend our proposed value of $\alpha$ and that it should be tuned the same way one would tune the temperature as the interpretation of $\alpha$ and $\tau$ is fairly similar, the higher $\alpha$, the higher entropy the policy will be. As for a recommended default, we chose $\alpha=0.77$ (obtained mathematically, not via hyperparameter tuning) to match SAC’s rule at the default scale (it was noted in the last paragraph of 8.2). This is why the performance of 4.a is almost identical to SAC’s.
>
> > The content in the experiment part lacks of sufficient details, which is also missing in the appendix. I recommend the authors to add the key details of experiments.
> >
>
> We have included all details of the experiments, including the experiment files (see `rexp.py` in the rl codebase) in the appendix. We do not understand which details are missing.
>
> > It seems that the coefficients $\gamma$ and $\tau$ are missing for the entropy term in Equation 3a.
>
> We added the missing $\tau$; thank you for noting this omission.
>
> > Is it possible to have some case-analysis for the results of DMC (e.g., in Figure 3,4)? For example, the proposed method achieves significantly better performance in tasks like BallInCupCatch, FingerTurnHard, HopperStand while the baseline fails totally.
> >
>
> We will include an ablation for the temperature in the static temperature. However, it might not be ready before the end of the discussion period.
>
> As for Figure 4.b, the scale was chosen such that the heuristic of SAC becomes infeasible, as noted in the last paragraph of 8.2.
>
> Do you have something particular in mind?
>
> > Which is the type of temperature setting adopted for the experiments in Section 8.3, i.e., fixed temperature or dynamic temperature?
> >
>
> We added a remark that the temperature is static. We assumed that the fact that we called it soft Q-learning (SQL) and not SAC would convey this information as the dynamic temperature was introduced with SAC, not SQL.

---

### Official Review · Reviewer_aWwr · 2023-11-09

**Soundness:** 2 fair
**Presentation:** 2 fair
**Contribution:** 2 fair
**Rating:** 5
**Confidence:** 3

**Summary:**

This paper focuses on regularized reinforcement learning, where the objective is to find a policy in an MDP which maximizes the reward minus a weighted regularization term—most commonly the negative entropy. The authors argue that the weight of the regularization term should vary depending on the environment and even sometimes based on the specific state, since the range of values the regularizer can take might depend heavily on the action space. To solve this problem, the authors introduce a method for dynamically setting the regularization coefficient based on the maximum and minimum values the regularizer can take at a particular state. They show that this can improve RL performance compared to using the same coefficient across environments.

**Strengths:**

As the authors note, regularized RL is widely applicable in both control and IRL, so improving its sensitivity to hyperparameters could be helpful for a variety of applications. The experimental results show that the proposed method for setting the regularization coefficient seems to work quite well in practice, particularly when the action space does not have a standard scale.

**Weaknesses:**

While the paper is promising, I worry that in it's current form it is not ready for publication at a top conference. First, the contribution is relatively small, since it is already known how to choose the entropy coefficient for many environments (e.g., via the SAC rule of using $\bar{H}=-n$) and it often needs to be tuned regardless depending on the scale of the reward function. While I still think the ideas here are useful, the writing, theory, and experiments should be of very high quality to justify publication.

However, I found the paper to be often unclear and imprecise, making it difficult to read. See the list of issues below for a partial list of the problems I noticed. The fact that many claims are unprecise and thus actually incorrect as stated makes it difficult to know which contributions to trust.

One possible extension to this work that could increase the contribution of the paper is to also include regularization to a base policy, as is often done in RLHF for LLMs via a KL divergence regularization term (e.g., see Bai et al., "Training a Helpful and Harmless Assistant with
Reinforcement Learning from Human Feedback"). There have been various papers attempting to find a systematic way of setting the regularization coefficient (e.g., the above paper and Gao et al., "Scaling Laws for Reward Model Overparameterization"), but none have considered possible setting the coefficient differently at different states, so this could be an interesting direction to apply the ideas presented in this paper.

Also, a relevant paper is "LESS is More: Rethinking Probabilistic Models of Human Behavior" by Bobu et al. They approach the problem from the motivation of modeling human behavior, which under the "Boltzmann rational" model is equivalent to solving a regularized MDP. They have a different approach to solving the issue of how the number of actions or number of trajectories that can achieve a particular reward affects the optimal regularized solution. It would be good to compare the approach in this paper with theirs.

**Issues with clarity and precision:**
 * The first paragraph of the intro argues "changing the action space should not change the optimal policy." This is true in most cases—the issue is that "changing the action space should not change the optimal *entropy-regularized* policy." I think it's important to make this distinction since otherwise it sounds wrong.
 * In Section 2, it is not specified that the MDPs considered have discrete actions, although it later appears this is an unspoken assumption. Then later, the paper switches to continuous MDPs without any explicit distinction.
 * In Section 2, according to the formalism in Geist et al., $\Omega$ should really be a function of only $\pi(s)$, not of $\pi$ in general.
 * In Section 2, the first time regularization is introduced it's with $\tau \Omega(\pi)$ subtracted from the value function; then immediately below it's instead added to the value function.
 * The definition of $\Omega^*_\tau$ is unclear.
 * Proof of Lemma 1: this proof seems insufficient; why exactly does convexity imply any other point must be smaller?
 * In SAC, the entropy is actually calculated before applying $\tanh$ to squash the actions into the action space. This means that the entropy is unbounded, and thus the idea in this paper does not apply directly.

**Typos:**
 * Proof of Lemma 1: "supermum" -> "supremum"
 * Section 3 title: "graviation" -> "gravitation"

**Questions:**

* Why should the ideas in this paper apply to SAC when the entropy as measured in SAC is actually unbounded?

---

> ### Author Response · Authors · 2023-11-16
>
> We are grateful for your time on our work and the report. We provide answers to the comments and questions below.
>
> > First, the contribution is relatively small, since it is already known how to choose the entropy coefficient for many environments (e.g., via the SAC rule of using
> $\bar{H}=-n$) and it often needs to be tuned regardless depending on the scale of the reward function.
> >
>
> The contribution of our work goes beyond a practical method for choosing the coefficient: we show that in problems with state-dependent action sets, our proposed solution improves the convergence conditions. This important theoretical and empirical finding is unknown in the literature. It is key to extending the scope of regularized methods in real-life problems beyond such as OR and drug discovery. In particular, in the drug discovery MDP, our proposed method is key to getting SQL to function properly. This approach had been discarded and shown to be (wrongly) inapplicable by the Gflownet authors.
>
> We agree that in MDPs where the actions are **not** state-dependent, we can tune the temperature for the environment. However, works like [1] consider one temperature for the whole benchmark, whereas our approach seamlessly adapts to each problem without the need for expensive hyperparameter tuning. Furthermore, as we show, the SAC rule (a constraint) need not be feasible, in which case the temperature will grow to infinity if it cannot be satisfied. Our proposed method would change the results in both cases.
>
> More importantly, we focus on a more general setting **with** state-dependent action sets where tuning the temperature will not be able to find our proposed temperature scheme, as the temperature proposed here depends on the number of actions. This was highlighted in the last experiment.
>
> Of course, one may argue that we can tune one temperature for every action size, but then the set of parameters to tune can become extremely large in MDPs like the drug design MDP (see last experiment).
>
> Lastly, we note that we are arguing for invariance with the scale of the actions, not the rewards.
>
> > The first paragraph of the intro argues "changing the action space
> should not change the optimal policy." This is true in most cases—the
> issue is that "changing the action space should not change the optimal *entropy-regularized* policy." I think it's important to make this distinction since otherwise it sounds wrong.
> >
>
> We changed it to “Changing the action space should not change the optimal regularized policy under the same changes” to rectify. Given that we work under the more general framework of regularized MDPs, the expression “*entropy-regularized* policy” is limiting.
>
> > In Section 2, it is not specified that the MDPs considered have discrete actions, although it later appears this is an unspoken assumption. Then later, the paper switches to continuous MDPs without any explicit distinction.
> >
>
> Our theory applies to both cases. In Section 4, we mention that “We look at differential entropy and continuous actions in Section 7.” We moved the phrase up so it’s more visible.
>
> > In Section 2, according to the formalism in Geist et al., $\Omega$ should really be a function of only $\pi(s)$, not of $\pi$ in general.
> >
>
> We denote the the regularized value as $\max_{\pi\in\Delta(\mathcal{A}(s))} \mathbb{E}_{a\sim\pi}[Q(s,a)] - \tau\Omega(\pi)$ where we denote the policy **at the current state** by $\pi$. This is accentuated by the fact that the domain of $\pi$ depends on the current state. In the spirit of being aligned with the notation of [2] and the aforementioned comment, we renamed the optimization variable to $\pi_s$
>
> > In Section 2, the first time regularization is introduced it's with subtracted from the value function; then immediately below it's instead added to the value function.
> >
>
> We fixed this typo; thank you for finding it.
>
> > The definition of $\Omega^*_\tau$is unclear.
> >
>
> Our definition follows [2]. See the equation
>
> $V(s) = \max_{\pi\in\Delta(\mathcal{A}(s))} \mathbb{E}_{a\sim\pi}[Q(s,a)] - \tau\Omega(\pi) = \Omega^\star_\tau(Q(s,\cdot)),$
>
> in Section 2. [2] define $\Omega^\star$ as
> $\Omega^*(q_s)=\max_{\pi_s\in\Delta_\mathcal{A}}\langle\pi_s,q_s\rangle-\Omega(\pi_s).$
>
> > Proof of Lemma 1: this proof seems insufficient; why exactly does convexity imply any other point must be smaller?
> >
>
> We added a clarification that we are looking at points between 0 and 1 as we are working with discrete distributions, which follows the notice in Section 4. Hence, this follows directly from strict convexity: $f(p)<pf(1) + (1 - p) f(0)$ for $0\leq p \leq 1$(see Section 4 of [3] in particular Theorem 4.1). Since we assumed that $f(1)=f(0)=0$ (see Assumption 1), $f(p)<p0+(1-p)0=0$.

---

> > ### Author Response · Authors · 2023-11-16
> >
> > > In SAC, the entropy is actually calculated before applying $tanh$ to squash the actions into the action space. This means that the entropy is unbounded, and thus the idea in this paper does not apply directly.
> >
> > Please see our answer to the first question.
> >
> > > **Questions:**
> > >
> > > - Why should the ideas in this paper apply to SAC when the entropy as measured in SAC is actually unbounded?
> >
> > Below, we provide ample evidence that not only the log prob is calculated after the tanh which makes the entropy bounded, but the entropy before the tanh is almost always bounded from above and below as the standard deviation is often clipped. If the entropy is measured before the tanh, then the action of selecting the highest movement can be represented by a very large average and high standard deviation which is odd. Lastly, the clipping of the temperature is the only component allowing SAC with the $-n$ rule to work in the experiment with scaled actions to [-0.1,0.1] as it would otherwise drift to infinity.
> >
> >
> >
> > [7, equation 20 in the supplementary] defines the policy as a Gaussian distribution with squashing.
> >
> > [The original SAC repository](https://github.com/haarnoja/sac) does the `tanh` correction for the log prob ([see here](https://github.com/haarnoja/sac/blob/8258e33633c7e37833cc39315891e77adfbe14b2/sac/policies/gaussian_policy.py#L74)) and also clamps the standard deviation ([see here](https://github.com/haarnoja/sac/blob/8258e33633c7e37833cc39315891e77adfbe14b2/sac/distributions/normal.py#L55)). Choosing a maximum standard deviation means that before the `tanh`, the maximum entropy is known by (see [“Specified mean and variance: the normal distribution”](https://en.wikipedia.org/wiki/Maximum_entropy_probability_distribution))
> >
> > [The updated version from RAIL](https://github.com/rail-berkeley/softlearning) calculates the entropy after the `tanh` see `action_distribution` [here](https://github.com/rail-berkeley/softlearning/blob/13cf187cc93d90f7c217ea2845067491c3c65464/softlearning/policies/gaussian_policy.py#L74) which is the bijective transformation defined [here](https://github.com/rail-berkeley/softlearning/blob/13cf187cc93d90f7c217ea2845067491c3c65464/softlearning/policies/base_policy.py#L267).
> >
> > [Spinning up](https://spinningup.openai.com/en/latest/), the RL tutorial from OpenAI uses the `tanh` log probs (see [here](https://github.com/openai/spinningup/blob/038665d62d569055401d91856abb287263096178/spinup/algos/pytorch/sac/core.py#L60) and the comment above it for an explanation) and clamps the standard deviation. (see [here](https://github.com/openai/spinningup/blob/038665d62d569055401d91856abb287263096178/spinup/algos/pytorch/sac/core.py#L42))
> >
> > [Cleanrl](https://github.com/vwxyzjn/cleanrl) [4] also includes both the [tanh log prob](https://github.com/vwxyzjn/cleanrl/blob/2d660b6d3053ea9037c746b4c9f3a6faa1f20c44/cleanrl/sac_continuous_action.py#L140) and the [clamped standard deviation](https://www.notion.so/Sobhan-ICLR-2024-Rebuttal-be69138c020f4243bdf65a7615dd7d0e?pvs=21).
> >
> > [Tianshou](https://github.com/thu-ml/tianshou) [5] also applies the l[og prob to the tanh](https://github.com/thu-ml/tianshou/blob/962c6d1e11d033b6faaa2da37a89794563baebf6/tianshou/policy/modelfree/sac.py#L168).
> >
> > Lastly, [6, equation 25] defines the policy with the action bound.
> >
> > > **Typos:**
> > >
> >
> > We corrected the typos; thank you for highlighting them.
> >
> > > Also, a relevant paper is "LESS is More: Rethinking Probabilistic Models of Human Behavior" by Bobu et al. They approach the problem from the motivation of modeling human behavior, which under the "Boltzmann rational" model is equivalent to solving a regularized MDP. They have a different approach to solving the issue of how the number of actions or number of trajectories that can achieve a particular reward affects the optimal regularized solution. It would be good to compare the approach in this paper with theirs.
> > >
> >
> > Thank you for the suggestion. However, we do not think they are directly comparable. First and foremost, LESS is for the inverse problem, not the forward one. Indeed, in the inverse problem (learning the reward function given observations, not a simulator), papers like [8] already estimate the optimal temperature per state.
> >
> > The second issue is that we do not assume any particular structure, while LESS assumes a reward that is linear in features. This is not learnable in the forward setting, as the linear feature may end up being the reward itself, defeating the whole purpose of LESS.
> >
> > A third issue is that it is unclear how LESS would be extensible to stochastic environments.
> >
> > While using trajectory similarity is interesting, LESS does not include a Bellman equation, and writing it seems nontrivial.

---

> > > ### Author Response · Authors · 2023-11-16
> > >
> > > > One possible extension to this work that could increase the contribution of the paper is to also include regularization to a base policy, as is often done in RLHF for LLMs via a KL divergence regularization term (e.g., see Bai et al., "Training a Helpful and Harmless Assistant with Reinforcement Learning from Human Feedback"). There have been various papers attempting to find a systematic way of setting the regularization coefficient (e.g., the above paper and Gao et al., "Scaling Laws for Reward Model Overparameterization"), but none have considered possible setting the coefficient differently at different states, so this could be an interesting direction to apply the ideas presented in this paper.
> > >
> > >
> > > We are grateful for the suggestion, but these suggestions would make the paper extremely long and are out of the scope of this work as they have a focus somewhat orthogonal to the focus of the paper.
> > >
> > >
> > > [1] Vieillard, Nino, Olivier Pietquin, and Matthieu Geist. "Munchausen reinforcement learning." *Advances in Neural Information Processing Systems* 33 (2020): 4235-4246.
> > >
> > > [2] Geist, Matthieu, Bruno Scherrer, and Olivier Pietquin. "A theory of regularized markov decision processes." *International Conference on Machine Learning*. PMLR, 2019.
> > >
> > > [3] Rockafellar, R. Tyrrell. *Convex analysis*. Vol. 11. Princeton university press, 1997.
> > >
> > > [4] Huang, Shengyi, et al. "Cleanrl: High-quality single-file implementations of deep reinforcement learning algorithms." *The Journal of Machine Learning Research* 23.1 (2022): 12585-12602.
> > >
> > > [5] Weng, Jiayi, et al. "Tianshou: A highly modularized deep reinforcement learning library." *The Journal of Machine Learning Research* 23.1 (2022): 12275-12280.
> > >
> > > [6] Haarnoja, Tuomas, et al. "Soft actor-critic algorithms and applications." *arXiv preprint arXiv:1812.05905* (2018).
> > >
> > > [7] Haarnoja, Tuomas, et al. "Soft actor-critic: Off-policy maximum entropy deep reinforcement learning with a stochastic actor." *International conference on machine learning*. PMLR, 2018.
> > >
> > > [8] Mai, Tien, Mogens Fosgerau, and Emma Frejinger. "A nested recursive logit model for route choice analysis." *Transportation Research Part B: Methodological* 75 (2015): 100-112.

---

> > ### Comment · Reviewer_aWwr · 2023-12-01
> > **Response to authors**
> >
> > Thank you for the comprehensive rebuttal. Most of my concerns are addressed and so I have raised my score. My remaining hesitations around acceptance are:
> >  * I still think the writing could be clearer and more precise in places.
> >  * Reviewer nhQG brought up some concerns around novelty and missing comparisons that haven't been addressed by the authors.

---

> ### Author Response · Authors · 2023-12-02
>
> Thank you for your feedback and for raising your evaluation score. We have addressed the concerns raised by nhQG and hope you find it interesting.

---

### Official Review · Reviewer_nhQG · 2023-11-22

**Soundness:** 3 good
**Presentation:** 3 good
**Contribution:** 2 fair
**Rating:** 6
**Confidence:** 3

**Summary:**

The paper proposes a new method of regularization of common RL algorithms. First, the paper proves that standard regularization techniques used in RL algorithms such as SQL and SAC suffer from regularization that behaves differently with the size of the action space. The authors propose a new family of regularization techniques where state-dependent temperature parameters are learned by normalizing by the range of regularization function. The authors also show that such normalization can also be used on regularization techniques that dynamically learn the temperature parameter. Finally, the authors demonstrate their results on a variety of domains, including DM env and drug discovery.

**Strengths:**

1. The paper motivates the problem with standard regularization techniques well in Section 2-5.

2. The empirical results are quite compelling. The proposed method, while simple, succeeds in various DM env tasks where standard methods fail. More convincingly, the proposed method allows SQL to succeed on a drug discovery task, whereas prior attempts were known to be too unstable.

**Weaknesses:**

1. The method proposed in the paper is very simple, and involves normalizing the standard temperature by the range the regularization objective can take (e.g. minimum possible entropy). While I do not think that the simplicity of the approach should detract from the novelty of the paper, I am not convinced that the better experimental results are actually due to the proposed range-normalization, and not simply that the regularization can now be state-dependent. To my knowledge, state-dependent temperatures is not itself a novel concept, as other works have tried to dynamically set temperature based on state [1, 2]. The results would be more convincing if the authors showed that prior efforts in state-dependent regularization fail whereas theirs succeeds.

2. The algorithm does not actually circumvent the extensive hyperparameter tuning required by existing RL algorithms. It is unclear how sensitive the proposed approach is to various settings of temperature and \alpha.


[1] https://proceedings.mlr.press/v70/asadi17a/asadi17a.pdf

[2] https://arxiv.org/pdf/2111.14204.pdf

**Questions:**

1. The empirical results swap between dynamic and static learning of the temperature parameter. Is there a reason why one of the method works better for a particular domain. Specifically, will dynamic temperature learning at least reproduce the results of using a static temperature in the drug discovery experiment?

---

> ### Author Response · Authors · 2023-12-02
>
> $\newcommand{\Af}{\unicode{x1D538}}$
> We are grateful for your time on our work and the report. We provide answers to the comments and questions below.
>
> > The method proposed in the paper is very simple, and involves normalizing the standard temperature by the range the regularization objective can take (e.g. minimum possible entropy). While I do not think that the simplicity of the approach should detract from the novelty of the paper, I am not convinced that the better experimental results are actually due to the proposed range-normalization, and not simply that the regularization can now be state-dependent. To my knowledge, state-dependent temperatures is not itself a novel concept, as other works have tried to dynamically set temperature based on state [1, 2]. The results would be more convincing if the authors showed that prior efforts in state-dependent regularization fail whereas theirs succeeds.
>
> We want first to note that we are not learning a per-state temperature; in all experiments (including DMC), the model's temperature is a single scalar, and the per-state temperature is that scalar divided by the range of the regularizer at the state. This means that in the DMC benchmark, the temperature is the same for all states for each environment. While our approach is more flexible as it effectively uses a different temperature for states with different action spaces, it differs from efforts in learning state-dependent gamma, lambda, and tau (e.g., [4]). While using a parametrized network that chooses the temperature is an interesting research venue, it is out of the scope of this work and not what we propose here.
>
> We want to separate the approaches of [1] and [2].
>
> [2] anneal the temperature as the state is visited more and more often. They effectively use temperature to both promote exploration and embed uncertainty. Indeed, the added temperature increases the likelihood of visiting that state as the value function increases, and the higher temperature increases the probability of taking nonoptimal actions.
>
> In many normal RL tasks, low temperature (almost zero) is desirable as we do not measure performance in a way where the added entropy is helpful. Indeed, the added entropy can be seen as a form of robustness in an adversarial environment (e.g., [3]). Yet, the test environment is often the same as the training and static, so the added entropy can hurt performance, making very small temperatures desirable. However, this insight does not transfer to the drug design MDP since our performance measure is not the best terminal state (drug) found. Instead, we measure the number of terminal states that are not similar to previously found terminal states; [6] refer to this as the number of modes. Indeed, setting the temperature too low hurts the performance. Of course, the approach of [2] is amendable as it can converge to a higher than zero temperature with minor modifications; it could even converge to our proposed temperature. In any case, [2] proposes something that does not change the final model but helps the agent learn it, making it orthogonal to our work.
>
> We are aware of [1] and cite it. We originally included a comparison [1] (referred to as mellowmax or MM) in an early draft in a similar but different setup. We decided not to include it in our comparisons as it worked too poorly; sampling with [1] requires solving a 1D optimization problem with bisection in the discrete case, and we are unaware of how [1]'s method can be extended to continuous control tasks. We also note that [1] only uses the dynamic temperature for the execution, not the value function evaluation. We believe this also sets apart our proposed solution with the work of [1]. Indeed, mellowmax could be used in parallel with what we propose here, and our method improves the performance of [1]. We are working on a section in the appendix comparing the performance of [1] and our method, but it could potentially not be ready before the end of the rebuttal period.

---

> ### Author Response · Authors · 2023-12-02
>
> Recall that [1] proposes using the mellowmax operator, defined as
> $$mm_\tau(Q(s, \cdot))=\tau\log\sum_{a\in\Af(s)}\exp(Q(s,a)/\tau),$$
> where $\Af(s)$ is the action space at state $s$. [5] show that mellowmax is the aggregation function of a regularized MDP where the regularization is the KL divergence with the uniform policy, i.e.
> $$
> V(Q(s,\cdot))=\max_{\pi_s\in\Delta(\Af(s))}\left[\sum_{a\in\Af(s)} Q(s,a)\pi_s(a) - \tau KL(\pi_s\|U_s)\right]=mm_\tau(Q(s,\cdot)),
> $$
> where $U_s$ is the uniform policy at state $s$. However
> $$-KL(\pi_s||U_s) = -\sum_{a\in\Af(s)}\pi_s(a)\log\left(\frac{\pi_s(a)}{ U_s(a)}\right)= -\sum_{a\in\Af(s)}\pi_s(a)(\log\pi_s(a) - \log U_s(a))$$ is equal to the entropy of $\pi_s$,
> $$H(\pi_s)=-\sum_a\pi_s(a)\log\pi_s(a)$$
> minus the maximum entropy
> $$H(U_s)=-\sum_{a\in\Af(s)}\frac{1}{|\Af(s)|}\log\frac{1}{|\Af(s)|}=-\log\frac{1}{|\Af(s)|}=-\sum_{a\in\Af(s)}\pi_s(a)\log\frac{1}{|\Af(s)|}.$$ The last step is true because $\sum_{a\in\Af(s)} \pi_s(a)=1$.
>
> Effectively, mellowmax is soft Q-learning with a penalty of $\tau \log|\Af(s)|$ per action. We leverage this insight to use path consistency with mellowmax as PCL has no log-sum-exp that we can replace with mellowmax.
>
>
> This added pessimism is harmful, especially in MDPs like the drug design MDP, where the agent can decide to terminate the episode early, and we cannot set the temperature to zero (or something very small). Most of the molecules SQL and GFNs find are from the longest possible trajectories in the MDP with nine fragments, but MM fails to reach that zone. This is problematic for MM. We believe this is the reason [1] propose finding an execution policy $\bar{\pi}$, such that $\sum_{a\in\Af(s)} \bar{\pi}(a|s)Q(s,a) = mm_\tau(Q(s,\cdot))$, i.e., the expected SARSA with policy $\bar{\pi}$ is equal to the value function. [1] show that the maximum entropy policy satisfying this constraint is in the form $sm_{\bar{\tau}}(Q(s,\cdot))$ where $sm_{\bar{\tau}}$ is the softmax function for some unknown temperature $\bar{\tau}$. As shown below, $\bar{\tau}$ is always higher than $\tau$, yet the added optimism (in form of higher temperature) does not undo pessimism.
>
> Using mellow-max is not helpful on DMC as the tasks do not have terminations. Thus, the Q function using mellowmax is equal to the Q function of SAC minus $\frac{H(U_s)}{1-\gamma}$, meaning that the optimal policy does not change as softmax is invariant to shifts and as mentioned before, we are not aware how we can use the dynamic temperature program of [1] in continuous control tasks.
>
> Proof that [1] always chooses a higher temperature when calculating $\bar{\pi}$: Since the KL divergence is always positive, $\max_{\pi_s} \pi_s(a)\sum_a Q(s,a) - \tau KL(\pi_s\|U_s)$ is always less than $\sum_a Q(s,a)\pi(a|s)$ where $\pi(\cdot|s)=sm_\tau(Q(s,\cdot))$ is the solution to the aforementioned maximization problem. On the other hand, the average, $\sum_a U_s(a)Q(s,a)$, is always a lower bound for the mellowmax as $U_s$ is a feasible solution in the maximization problem and also corresponds to the policy when the temperature is $\infty$. [1] Show that $f(\hat{\tau}) =\sum_a sm_{\hat{\tau}}(Q(s,\cdot))(a)Q(s,a) - mm_\tau(Q(s,\cdot))$ is monotonously decreasing with respect to $\hat{\tau}$ so the root is greater than or equal to $\tau$.

---

> ### Author Response · Authors · 2023-12-02
>
> > The algorithm does not actually circumvent the extensive hyperparameter tuning required by existing RL algorithms. It is unclear how sensitive the proposed approach is to various settings of temperature and \alpha.
>
> The sensitivity of the approach proposed here is not different from other methods save for one major difference: our proposed heuristic never becomes infeasible. In the state-dependent action setting, the maximum entropy a policy can have depends on the states it visits, not just the final policy, so while a target could be reasonable for the final policy, it could be infeasible for the initial policy. In the state-independent action setting, the target has to be carefully chosen to be below the maximum entropy, which is one of the things we propose. If the SAC temperature program becomes infeasible, the temperature will diverge to infinity, leading to the agent doing a random walk, which can be problematic.
>
> Our approach shines where one temperature has to be chosen for a wide range of tasks, such as benchmarks or in the state-dependent action setting. In both cases, the number of temperatures that need choosing, if choosing one temperature per action space or environment, can be too high to be searchable or not finite.
>
> Lastly, we’d like to refer to the relaxed sufficient conditions we propose for undiscounted entropy regularized MDPs. This cannot be achieved with hyperparameter tuning unless one temperature is chosen per action space.
>
> > 1. The empirical results swap between dynamic and static learning of
> the temperature parameter. Is there a reason why one of the method works better for a particular domain. Specifically, will dynamic temperature learning at least reproduce the results of using a static temperature in the drug discovery experiment?
>
> We want to start by noting that we used the standard setup, and the jump you mentioned is pure coincidence. We used dynamic temperature for DMC because this is what we found to be the baseline. We use the static temperature for DMC before the dynamic part because it is less common. Lastly, we use static temperature when comparing the GFNs because we want to keep the implementation as similar as possible to GFNs.
>
> The approaches are compatible in that for temperature $\tau$ and a policy $\pi$ being the solution of the soft Bellman equation with temperature $\tau/H(U_s)$ at state $s$, there exists heuristic parameter $\alpha$ such that the dynamic temperature program of SAC does not change the temperature.
>
> In short, we look at the gradient of
> $$
> \mathbb{E}_s\left[\frac{\tau}{H(U_s)}\left(H(\pi(\cdot|a)) - \alpha H(U_s)\right)\right]
> $$
> with respect to $\tau$ and show that there exists a value of $\alpha$ such that the gradient is zero. The aforementioned equation is the objective we get for the temperature program of SAC when combining the two approaches we propose: a temperature that depends on the state, not with a neural network, but by scaling using the range of the regularizer and a target that is a ratio of the maximum regularizer.
>
> The gradient is
> $$
> \mathbb{E}_s\left[\frac{H(\pi(\cdot|a))}{H(U_s)} - \alpha \right]
> $$
>
> Which can be zero if we set $\alpha$ to $\mathbb{E}_s[H(\pi)/H(U_s)]$ as $\mathbb{E}_s[H(\pi)/H(U_s)]$ exists and is always non-negative and less than or equal to one as $H(U_s)$ is an upper bound for $H(\pi(\cdot|a))$.
>
> While there is an equivalent constant target equal to $\alpha\mathbb{E}_s[H(U_s)]$ under the final policy, it does **not** need to be feasible while training the model.
>
> [1] Asadi, Kavosh, and Michael L. Littman. "An alternative softmax operator for reinforcement learning." International Conference on Machine Learning. PMLR, 2017.
>
> [2] Hu, Dailin, Pieter Abbeel, and Roy Fox. "Count-Based Temperature Scheduling for Maximum Entropy Reinforcement Learning." arXiv preprint arXiv:2111.14204 (2021).
>
> [3] Derman, Esther, Matthieu Geist, and Shie Mannor. "Twice regularized MDPs and the equivalence between robustness and regularization." *Advances in Neural Information Processing Systems* 34 (2021): 22274-22287.
>
> [4] Zhao, Mingde, et al. "META-Learning State-based Eligibility Traces for More Sample-Efficient Policy Evaluation." arXiv preprint arXiv:1904.11439 (2019).
>
> [5] Geist, Matthieu, Bruno Scherrer, and Olivier Pietquin. "A theory of regularized markov decision processes." International Conference on Machine Learning. PMLR, 2019.
>
> [6] Bengio, Emmanuel, et al. "Flow network based generative models for non-iterative diverse candidate generation." Advances in Neural Information Processing Systems 34 (2021): 27381-27394.

---

### Meta-Review · Area_Chair_MK9R · 2023-12-20

**Metareview:**

The paper addresses the issue of standard regularization techniques in reinforcement learning varying with action space size by introducing state-dependent temperature parameters. Reviewers appreciated the method's motivation and simplicity, as well as its analysis and empirical evaluation in domains including biological design. Concerns were raised about the size of the paper's contribution given the scope of the approach, and continuing reliance on hyperparameter tuning. The reviewers nonetheless believed the paper presents an interesting and well-motivated tool to the community and for this reason I recommend acceptance.

**Justification For Why Not Higher Score:**

Limited scope of impact.

**Justification For Why Not Lower Score:**

Highlights issues with existing approaches, and is simple.

---

### Decision · Program_Chairs · 2024-01-16

Accept (poster)